# Vehicle Spatial Distribution and 3D Trajectory Extraction Algorithm in a Cross-Camera Traffic Scene

**DOI:** 10.3390/s20226517

**Published:** 2020-11-14

**Authors:** Xinyao Tang, Huansheng Song, Wei Wang, Yanni Yang

**Affiliations:** School of Information Engineering, Chang’an University, Xi’an 710064, China; andy19966212@126.com (X.T.); wangwei_211@chd.edu.cn (W.W.); yanniyang@chd.edu.cn (Y.Y.)

**Keywords:** camera calibration, cross-camera traffic scene, road panoramic image, vehicle spatial distribution, 3D trajectory extraction

## Abstract

The three-dimensional trajectory data of vehicles have important practical meaning for traffic behavior analysis. To solve the problems of narrow visual angle in single-camera scenes and lack of continuous trajectories in 3D space by current cross-camera trajectory extraction methods, we propose an algorithm of vehicle spatial distribution and 3D trajectory extraction in this paper. First, a panoramic image of a road with spatial information is generated based on camera calibration, which is used to convert cross-camera perspectives into 3D physical space. Then, we choose YOLOv4 to obtain 2D bounding boxes of vehicles in cross-camera scenes. Based on the above information, 3D bounding boxes around vehicles are built with geometric constraints which are used to obtain projection centroids of vehicles. Finally, by calculating the spatial distribution of projection centroids in the panoramic image, 3D trajectories of vehicles are extracted. The experimental results indicate that our algorithm can effectively complete vehicle spatial distribution and 3D trajectory extraction in various traffic scenes, which outperforms other comparison algorithms.

## 1. Introduction

Vehicle spatial distribution and 3D trajectory extraction is an important sub-task in the field of computer vision. With the development of intelligent transportation systems (ITS), a large amount of vehicle trajectory data reflecting movements is obtained through traffic surveillance videos, which can be used for traffic behavior analysis [1,2] such as speeding and lane change, traffic flow parameter (volume, density, etc.) calculation and prediction [3,4,5] and so on. Based on these data, traffic state estimation [6,7] and traffic management and control [8] can be conducted, which plays a key role in ensuring traffic efficiency and is of great research significance and practical value.

In current applications, trajectories mainly refer to two-dimensional trajectories in the image space, which do not contain spatial information of vehicles in the real world. Compared with 2D trajectories, 3D trajectories have one more dimension of spatial information, which has more obvious advantages in practical applications and can be further applied to traffic accident scene reconstruction and responsibility identification [9], as well as vehicle path planning [10] in autonomous driving and cooperative vehicle infrastructure system (CVIS) to avoid collision.

Currently, the most commonly used methods for obtaining 3D vehicle trajectories are based on object detection and feature point methods [11,12,13], which have been maturely applied in single-camera scenes. With the development of deep convolutional neural networks (DCNNs), several excellent object detection networks [14,15,16,17,18] have emerged, which greatly improve the accuracy and speed of object detection compared with the traditional feature extraction and classifier methods [19]. Based on object detection, feature points are extracted for vehicles to obtain 3D trajectories in the world space combined with camera calibration. Although these methods have been widely and maturely used in single-camera scenes, the trajectory results are not accurate under the condition of low camera perspectives and vehicle occlusion. To solve the problem, 3D object detection is considered because only the presence, 2D location and rough type of vehicles in the image space can be obtained by 2D object detection method and it is difficult to achieve a fine-grained description of vehicles. Compared with 2D methods, perspective distortion can be eliminated by 3D object detection. Moreover, 3D bounding box fits the vehicle better and can describe vehicle size, pose, and other information on the physical scale. Therefore, 3D model is more suitable for 3D trajectory extraction in traffic scenes. At the same time, the visual angle of single-camera scene is usually narrow, which is unable to meet the needs of applications in wide range scenes, so it is necessary to solve the problem of 3D vehicle trajectory extraction in the whole space.

At present, full space fusion mainly relies on multi-scene stitching methods, which can be divided into two categories. (1) Image stitching based on image alignment [20,21]. The feature points of the overlapping areas in multiple images are detected and matched to construct homography matrixes between images. Then, the panoramic image is generated based on the matrixes. This kind of method is mature and widely used, especially in the panoramic photography of mobile phone applications [22]. However, camera calibration is not used in these methods, which means physical information cannot be reflected in the panoramic image. (2) Image stitching based on camera calibration [23,24]. The transformation between world coordinate systems of the scenes are determined based on overlapping areas in the image and camera calibration to generate the panoramic image which contains actual physical information and can be used to measure and locate the world coordinates in the image. However, this kind of method requires complicated manual calibration of each camera, and has rarely been used in large scope of road measurement.

Methods of vehicle trajectory extraction in the whole space is cross-camera vehicle tracking, which means obtaining continuous vehicle trajectory from images taken by multiple cameras with or without overlapping areas. These methods usually contain three essential steps: camera calibration, vehicle detection, and tracking in single-camera scenes and cross-camera vehicle matching. For cameras with overlapping areas, the spatial correlation can be calculated by overlapping areas to obtain continuous trajectories. However, in practical applications, “blind areas” are often existed in images taken by multiple cameras. In case of this condition, methods of re-identification are used to accurately and efficiently match vehicles in different perspectives through vehicle apparent features. Then, continuous vehicle trajectories in the whole space can be obtained by space-time information inference.

Currently, re-identification methods used in cross-camera vehicle tracking are mostly based on vehicle features, such as vehicle color, shape, and texture, among which SIFT feature is the most commonly used due to its invariance to light, rotation, and scale. However, robustness of SIFT to affine transformation is low. To improve this problem, Hsu et al. [25] proposed a method of cross-camera vehicle matching based on ASIFT feature and min-hash technique, which can overcome the influence of multi-camera perspectives to feature detection but cannot obtain 3D vehicle trajectory and solve the problem of vehicle occlusion. Castaneda et al. [26] proposed a method of multi-camera detection and tracking of vehicles in non-overlapping tunnel scene, which uses optical flow and Kalman filter for vehicle tracking and state estimation in single scenes. Due to the special light environment in tunnel scene, vehicle color cannot be used as matching criterion. Thus, vertical and horizontal signatures are proposed to describe the similarity between vehicles. Combined with cross-camera vehicle travel time and lane position constraints, continuous vehicle trajectory can be obtained. To some extent, the problem of vehicle occlusion can be solved in this method, but the physical location of vehicle trajectory in 3D space is still not available. To further obtain vehicle trajectory in 3D space, multi-scene cameras should be calibrated in advance and the topological relationship between cameras should be determined to convert multi-camera perspectives into point sets in 3D coordinate system. Straw et al. [27] proposed a method of cross-camera vehicle tracking which uses DLT and triangulation for camera calibration and Kalman filter for vehicle state estimation. Although continuous trajectory in 3D space could be obtained, the accuracy is low, which cannot meet practical applications. Peng et al. [28] proposed a method of multi-camera vehicle detection and tracking in non-overlapping traffic surveillance, using convolutional neural network (CNN) for object detection and feature extraction and homography matrix for displaying vehicle trajectory to satellite map. This method can accurately show vehicle trajectory in panoramic map, but these trajectories do not contain physical location in 3D space. Byeon et al. [29] proposed an online method of cross-camera vehicle positioning and tracking, which uses Tsai two-step calibration method for camera calibration and represents vehicle matching as multi-dimensional assignment to solve the problem of vehicle matching in multi-camera scenes. Vehicle trajectory can be obtained in this method, but the road panoramic image with spatial information is not generated. Qian et al. [30] proposed a cross-camera vehicle tracking system for smart cities which uses object detection, segmentation, and multi-object tracking algorithms to extract vehicle trajectories in single-camera scenes. Then, a cross-camera multi-object tracking network is proposed to predict a matrix which measures the feature distance between trajectories in single-camera scenes. The system won the first place in AI City 2020 Challenge and can better solve the problem of vehicle matching in cross-camera scenes. However, continuous 3D trajectories of vehicles and the panoramic image of the scene cannot be obtained.

In view of the problems existing in current cross-camera vehicle tracking methods, such as the influence of visual angle, vehicle occlusion, and lack of continuous trajectories in 3D space, we propose an algorithm of vehicle spatial distribution and 3D trajectory extraction in cross-camera traffic scene. The main contributions of this paper are summarized as follows:A method of road space fusion in cross-camera scenes based on camera calibration is proposed to generate a road panoramic image with physical information, which is used to convert cross-camera perspectives into 3D physical space.A method of 3D vehicle detection based on geometric constraints is proposed to accurately obtain projection centroids of vehicles, which is used to describe vehicle spatial distribution in the panoramic image and 3D trajectory extraction of vehicles.

The rest of this paper is organized as follows. The proposed algorithm to complete vehicle spatial distribution and 3D trajectory extraction is illustrated in Section 2. Experiment results and some comparison experiments are presented in Section 3. Conclusions and future work are given out in Section 4.

## 2. Materials and Methods

### 2.1. Framework

The overall flow chart of the proposed algorithm is shown in Figure 1. First, a panoramic image of the road with spatial information is generated based on camera calibration, which is used to convert the cross-camera perspective into 3D physical space. Secondly, 3D bounding box is constructed by geometric constraints, which is used to obtain the projection centroid of the vehicle. Finally, 3D trajectory of the vehicle is extracted by calculating the spatial distribution of the projection centroid in the road panoramic image.

### 2.2. Road Space Fusion in Cross-Camera Scene

#### 2.2.1. Camera Calibration Model and Parameter Calculation

To complete road space fusion in cross-camera scenes, the relationship between 2D image space and 3D world space must be derived through camera calibration. In this paper, we refer to the study [31] and our previous work [32,33] to define coordinate systems and camera calibration model, and choose the single vanishing point-based calibration method VWL (One Vanishing Point, Known Width and Length) to complete the calculation of calibration parameters.

Schematic diagram of coordinate system and camera calibration model is shown in Figure 2. In this paper, three coordinate systems are defined, all of which are right-handed. The world coordinate system is defined by the x, y, z axis, and the origin $O_{w}$ is located at the projection point of the camera on the road plane, whereas z is perpendicular to the road plane upwards. The camera coordinate system is defined by the $x_{c},\;y_{c},\;z_{c}$ axis, and the origin $O_{c}$ is located at the camera optical center, and $x_{c}$ is parallel to x, $z_{c}$ pointing to the ground along the camera optical axis, $y_{c}$ perpendicular to the plane $x_{c}O_{c}z_{c}$. The image coordinate system is defined by u, v axis, and the origin $O_{i}$ is located at image center. In the image coordinate system, u is horizontal right and v is vertical downward. $z_{c}$ intersects the road plane at $r=(c_{x},c_{y})$ in the image coordinate system, which is called the principal point and its default location is at the center of the image. $c_{x},c_{y}$ represent half of the image width and height, respectively.

In camera calibration, calibration parameters usually include camera focal length f, camera height h above the road plane, tilt angle ϕ and pan angle θ. In addition, roll angle can be represented by a simple image rotation, which has no effect on calibration results and is not considered in this paper. Through the camera model, the projection expression from the world coordinate system to the image coordinate system can be deduced as follows:(1)$$\alpha \left[\begin{array}{c} u\\ v\\ 1 \end{array}\right]=\left[\begin{array}{rrrr} f & 0 & 0 & 0\\ 0 & -f\sin\phi & -f\cos\phi & fh\cos\phi \\ 0 & \cos\phi & -\sin\phi & h\sin\phi \end{array}\right]\left[\begin{array}{c} x\\ y\\ z\\ 1 \end{array}\right],$$
where α≠0 is the scale factor, the homogeneous coordinates of the world point and its projection are ${\left[\begin{array}{cccc} x & y & z & 1 \end{array}\right]}^{T}$ and ${\left[\begin{array}{ccc} u & v & 1 \end{array}\right]}^{T}$.

In this paper, the single vanishing point-based calibration method VWL [31,32] is adopted to solve the calibration parameters f,h,ϕ,θ, and the vanishing point $VP=(u_{0},v_{0})$ along the direction of traffic flow is extracted by road edge lines.

As shown in Figure 3, a line segment in the world coordinate system and its projection in the image coordinate system are presented, respectively. In Figure 3a, due to the pan angle θ, the point at infinity along the road direction can be expressed as $x_{0}={\left[\begin{array}{cccc} -\tan\theta & 1 & 0 & 0 \end{array}\right]}^{T}$ in the world homogeneous coordinate. In Figure 3b, according to the vanishing point principle, $\left(u_{0},\;v_{0}\right)$ is the projection of $x_{0}$ in the image coordinate system. From Equation (1), the calibration parameters ϕ,θ can be solved as follows:(2)$$\phi =\arctan(-v_{0}/f),$$
(3)$$\theta =\arctan(-u_{0}\cos\phi /f),$$

Besides vanishing points, markings on the road plane are also commonly used signs. In Figure 3a, the physical length of a line segment parallel to the road direction is l. The vertical coordinates of the front and back point are $y_{b}$, $y_{f}$ and $v_{b}$, $v_{f}$, where y represents the world coordinate system while v the image. The physical width of the road is w with a pixel length δ in the corresponding image coordinate system. It can be obtained from literature [31] that h can be expressed by w or l indirectly as follows:(4)$$h=\frac{fw\sin\phi }{\delta \cos\theta },$$
(5)$$h=\frac{f\tau l\cos\phi }{\;f^{2}+v_{0}{}^{2}},$$
where $\tau =\left(v_{f}-v_{0})(v_{b}-v_{0}\right)/\left(v_{f}-v_{b}\right)$, sinϕ,cosϕ,cosθ can be solved from Equation (2) and (3). By equating Equations (4) and (5) and substituting into sinϕ,cosϕ,cosθ, a fourth-order equation in f can be derived as:(6)$$f^{4}+[2(u_{0}{}^{2}+v_{0}{}^{2})-k_{V}{}^{2}]f^{2}+{\left(u_{0}{}^{2}+v_{0}{}^{2}\right)}^{2}-k_{V}{}^{2}v_{0}{}^{2}=0$$
where $k_{V}=\delta \tau l/\left(wv_{0}\right)$.

From Equation (6), f can be solved first. When f is uniquely determined, ϕ,θ can be solved according to Equations (2) and (3), and h can be finally solved according to Equations (4) or (5). Thus, all the calibration parameters are calculated and the mapping between world and image can be described according to Equation (1).

To illustrate the road space in a straightforward way, the origin of the image coordinate system $O_{i}$ and the y axis of the world coordinate system are adjusted. First, the origin of the image coordinate system is moved to the upper left corner of the image, corresponding to the change of the internal parameter matrix K:$$K=\left[\begin{array}{ccc} f & 0 & c_{x}\\ 0 & f & c_{y}\\ 0 & 0 & 1 \end{array}\right]$$

Then, the y axis is adjusted to the direction along the traffic flow. Therefore, the rotation matrix R contains two parts, respectively representing a rotation of ϕ+π/2 about the x axis and θ about the z axis, which can be specifically expressed as:$$R=R_{x}(\phi +\pi /2)R_{z}(\theta )=\left[\begin{array}{rrr} \cos\theta & -\sin\theta & 0\\ -\sin\phi \sin\theta & -\sin\phi \cos\theta & -\cos\phi \\ \cos\phi \sin\theta & \cos\phi \cos\theta & -\sin\phi \end{array}\right]$$

The translation matrix is:$$T=\left[\begin{array}{cccc} 1 & 0 & 0 & 0\\ 0 & 1 & 0 & 0\\ 0 & 0 & 1 & -h \end{array}\right]$$

Therefore, the adjusted mapping from world point (x,y,z) to image point (u,v) in homogeneous form can be expressed as:$$s\left[\begin{array}{c} u\\ v\\ 1 \end{array}\right]=KRT\left[\begin{array}{c} x\\ y\\ z\\ 1 \end{array}\right]=H\left[\begin{array}{c} x\\ y\\ z\\ 1 \end{array}\right]$$
where $H=\left[h_{ij}\right],i=1,2,3;j=1,2,3,4$ is the 3×4 projection matrix from the world coordinate to the image coordinate, and s is the scale factor.

Finally, according to the derivation, the adjusted mapping between world and image can be described as follows:(7)$$\mathrm{World}-\mathrm{to}-\mathrm{Image}\;\left\{ \begin{array}{l} u=\frac{h_{11}x+h_{12}y+h_{13}z+h_{14}}{h_{31}x+h_{32}y+h_{33}z+h_{34}}\\ v=\frac{h_{21}x+h_{22}y+h_{23}z+h_{24}}{h_{31}x+h_{32}y+h_{33}z+h_{34}} \end{array}\right.,$$
(8)$$\mathrm{Image}-\mathrm{to}-\mathrm{World}\;\left\{ \begin{array}{l} x=\frac{b_{1}(h_{22}-h_{32}v)-b_{2}(h_{12}-h_{32}u)}{\left(h_{11}-h_{31}u)(h_{22}-h_{32}v)-(h_{12}-h_{32}u)(h_{21}-h_{31}v\right)}\\ y=\frac{-b_{1}(h_{21}-h_{31}v)+b_{2}(h_{11}-h_{31}u)}{\left(h_{11}-h_{31}u)(h_{22}-h_{32}v)-(h_{12}-h_{32}u)(h_{21}-h_{31}v\right)} \end{array}\right.,$$
where $\left\{ \begin{array}{l} b_{1}=u(h_{33}z+h_{34})-(h_{13}z+h_{14})\\ b_{2}=v(h_{33}z+h_{34})-(h_{23}z+h_{24}) \end{array}\right.$.

#### 2.2.2. Unified World Coordinate System and Road Panoramic Image Generation

The mapping between world and image in a single scene can be described through camera calibration. To complete 3D vehicle trajectory extraction in cross-camera scenes, the road space needs to be fused. At present, image stitching methods are often used, but most of them rely on overlapping areas to extract feature points for matching and obtaining transformation of scenes. However, feature extraction and matching are time-consuming. For multi-scene (more than two scenes) stitching, accumulated errors are existed in transformation of scenes, which will affect the quality of final image stitching result and the measurement accuracy of physical distance. Therefore, we propose a road space fusion algorithm in cross-camera scenes based on camera calibration which is not completely dependent on overlapping areas between scenes. When there are no overlapping areas between scenes, only the distances between cameras are needed.

Schematic diagram of road space fusion in cross-camera scenes is shown in Figure 4. In Figure 4a, number of cameras in the scene is N(N≥2), the set of sub-scene world coordinate systems is defined as $\left\{W_{s}^{i}:O_{w}^{i}-x_{i}y_{i}z_{i};i=1,2,\cdots ,N\right\}$, which is the same as the world coordinate system in the single scene described in the previous section. The unified world coordinate system is defined as $W_{u}:O_{u}-x_{u}y_{u}z_{u}$, and the origin $O_{u}$ is located in the road edge close to the camera. $O_{u}O_{w}^{1}$ is perpendicular to the road edge. The mapping matrix between the world coordinate system and the image coordinate system of each scene is the adjusted result described in the previous section, which is defined as $H_{i},i=1,2,\cdots ,N$. The red dots in Figure 4 are the control points set to identify the road areas. Two control points are set for each scene. The sets of control points in image and world coordinate system are $\left\{P_{2d}^{i}:p_{1}^{i},p_{2}^{i};i=1,2,\cdots ,N\right\}$ and $\left\{P_{3d}^{i}:P_{1}^{i},P_{2}^{i};i=1,2,\cdots ,N\right\}$ respectively. In Figure 4b, the panoramic image coordinate system is defined as $O_{p}-u_{p}v_{p}$, and the origin $O_{p}$ is located at the upper left corner of the panoramic image, which is similar to the image coordinate system.

Schematic diagram of road distribution in the panoramic image is shown in Figure 5. The proposed road space fusion algorithm in cross-camera scenes is specifically illustrated with this figure.
Step 1:Camera calibration. The calibration method proposed in this paper is used to calculate calibration parameters of each camera in the scene, including internal parameter matrix $K_{i}$, rotation matrix $R_{i}$, translation matrix $T_{i}$ and projection matrix of each camera $H_{i}=K_{i}R_{i}T_{i};i=1,2,\cdots ,N$.Step 2:Road area identification by setting control points. Harris corner extraction algorithm is used to obtain the image coordinate set of the nearest and furthest marking endpoints on the road plane in each scene, which is denoted as $\left\{P_{2d}^{i}:p_{1}^{i}=(x_{1}^{i},y_{1}^{i}),p_{2}^{i}=(x_{2}^{i},y_{2}^{i});i=1,2,\cdots ,N\right\}$. Equation (8) is used to convert $P_{2d}^{i}$ to the world coordinate set $\left\{P_{3d}^{i}:P_{1}^{i}=(X_{1}^{i},Y_{1}^{i},0),P_{2}^{i}=(X_{2}^{i},Y_{2}^{i},0);i=1,2,\cdots ,N\right\}$. The range of road area is calculated from $P_{3d}^{i}$ as $\left\{R_{l}^{i}:\left|Y_{2}^{i}-Y_{1}^{i}\right|;i=1,2,\cdots ,N\right\}$.Step 3:Set control parameter groups and divide pixels of the panoramic image $M_{p}$ into corresponding scenes. The width of the road is w (mm). The scale in the road space along the width direction is $r_{w}$ (pixel/mm) and the length direction $r_{l}$. The height and width of $M_{p}$ are $wr_{w}$ and $r_{l}\sum_{i=1}^{N}\left|Y_{2}^{i}-Y_{1}^{i}\right|$, where the corresponding length of each scene on the panoramic image $M_{p}$ is $r_{l}\left|Y_{2}^{i}-Y_{1}^{i}\right|;i=1,2,\cdots ,N$.Step 4:Generate the complete panoramic image $M_{p}$. The panoramic image coordinates are traversed from the origin at the upper left corner. A point (u,v) in the image coordinate system belongs to scene i and its corresponding world coordinate point is $\left(X_{1}^{i}+v/r_{w}-w/2,Y_{1}^{i}+\left(u-R_{i}\right)/r_{l},0\right)$, where $R_{i}=\{\begin{array}{ll} 0 & i=1\\ \sum_{t=1}^{i-1}r_{l}\left|Y_{2}^{t}-Y_{1}^{t}\right| & i=2,3,\cdots ,N \end{array}$.

The pixel in the road area $I_{pixel}$ corresponding to the world coordinate point is taken out (if any) and put to the position of the panoramic image coordinate point. Repeat this process until all the pixels of the corresponding road areas in all scenes are taken out and put into the panoramic image correctly.

Since the generated panoramic image contains physical information of road space, the position in the sub-scene world coordinate system and the unified world coordinate system can be calculated directly from a point in the panoramic image. In addition, the position in the unified world coordinate system and the panoramic image coordinate system can also be analyzed from a point in the sub-scene world coordinate system. The specific mapping equation group is as follows:panoramic image-to-world
(9)$$\{\begin{array}{ll} Unified\;world\;coordinate & \left(v/r_{w}+X_{1}^{i}-w/2,u/r_{l}+Y_{1}^{1},0\right)\\ Subscene\;world\;coordinates & \left(v/r_{w}+X_{1}^{i}-w/2,(u-R_{i}\right)/r_{l}+Y_{1}^{i},0) \end{array},$$
where a point in the panoramic image is denoted as (u,v), i represents the number of the sub-scene, $R_{i}=\{\begin{array}{ll} 0 & i=1\\ \sum_{t=1}^{i-1}r_{l}\left|Y_{2}^{t}-Y_{1}^{t}\right| & i=2,3,\cdots ,N \end{array}$.
Sub-scene world-to-panoramic image
(10)$$\begin{array}{cc} panoramic\;image\;coordinate & \left(r_{l}(Y-Y_{1}^{i}+U_{i}),r_{w}[X-(X_{1}^{i}-w/2)]\right) \end{array},$$where a point in sub-scene i is denoted as (X,Y,0), $U_{i}=\{\begin{array}{ll} 0 & i=1\\ \sum_{t=1}^{i-1}\left|Y_{2}^{t}-Y_{1}^{t}\right| & i=2,3,\cdots ,N \end{array}$.

### 2.3. 3D Vehicle Detection for Distribution and Trajetory Extraction

#### 2.3.1. 3D Bounding Boxes and Projection Centroids of Vehicles

Based on road space fusion in cross-camera scenes, to further obtain vehicle spatial distribution and 3D trajectory, vehicle detection in the scene is needed. Since the height of vehicle feature points is unknown, projection centroid is adopted in this paper instead, which depends on 3D vehicle detection. Considering actual application requirements, we choose YOLOv4 [34] for 2D vehicle detection. The detection results contain center point, width, and height of 2D bounding box in the image coordinate system, vehicle type (car, truck, bus) and its confidence. Then, the best 3D vehicle detection result and projection centroid are obtained by geometric constraints for vehicle spatial distribution and 3D trajectory extraction.

Figure 6 shows the vehicle model of 2D/3D bounding box from left and right perspectives. In each sub-figure, the left represents 2D model while the right 3D model. 2D model is in the image coordinate system. The axes in 3D model are the same direction as the world coordinate system, and the origin is the bottom left point of the 3D model. The vertices of 2D bounding box model are numbered from 0 to 3, and the corresponding image coordinates are denoted as $P_{i}^{2D}=(u_{i}^{2D},v_{i}^{2D}),i=0,1,2,3$. In the same way, the vertices of 3D bounding box model are numbered from 0 to 7, and the corresponding world and image coordinates are denoted as $P_{i}^{3D}=(x_{i}^{3D},y_{i}^{3D},z_{i}^{3D}),i=0,1,\cdots ,7$ and $P_{j}^{3\mathrm{Di}}=(u_{j}^{3\mathrm{Di}},v_{j}^{3\mathrm{Di}}),j=0,1,\cdots ,7$. The world coordinates of eight vertices and projection centroid of the vehicle from different perspectives are presented in Table 1.

Schematic diagram of 2D/3D vehicle detection is shown in Figure 7 (the left represents 2D detection while the right 3D detection) and the algorithm is specifically described as follows:
Step 1:YOLOv4 is used to obtain the vertices in the image coordinate system $P_{i}^{2D}=(u_{i}^{2D},v_{i}^{2D}),i=0,1,2,3$ and vehicle type. The base point of 2D bounding box is set as $P_{1}^{2D}=(u_{1}^{2D},v_{1}^{2D})$ in the image coordinate system, which can be converted into $P_{1}^{3D}=(x_{1}^{3D},y_{1}^{3D},z_{1}^{3D})$ in the world coordinate system by Equation (8), where $z_{1}^{3D}=0$.Step 2:Suppose 3D vehicle physical size $\left(l_{v},w_{v},h_{v}\right)$, $l_{v}$,$w_{v}$,$h_{v}$ represent vehicle length, width, and height respectively. According to Table 1, the world coordinates of the eight vertices in 3D bounding box model are calculated as $P_{i}^{3D}=(x_{i}^{3D},y_{i}^{3D},z_{i}^{3D}),i=0,1,\cdots ,7$.Step 3:The calculation results in Step 2 are converted to the image coordinates $P_{j}^{3\mathrm{Di}}=(u_{j}^{3\mathrm{Di}},v_{j}^{3\mathrm{Di}}),j=0,1,\cdots ,7$ through Equation (7) to complete 3D vehicle detection.

#### 2.3.2. Geometric Constraints

According to the above 3D vehicle detection algorithm, obtaining accurate 3D vehicle physical size is the premise to complete precise 3D vehicle detection. Due to the factor of perspective distortion and lack of depth information in monocular image, accurate size cannot be obtained by vehicle type which is derived from YOLOv4. Therefore, geometric constraints are considered to accurately calculate 3D vehicle physical size, which includes diagonal constraint and vanishing point constraint.

3D vehicle detection is equivalent to obtaining 3D vehicle physical size $X=(l_{v},w_{v},h_{v})$, and the diagonal pixel length of 2D bounding box is defined as:(11)$$l_{2D}={\left\|P_{1}^{2D}-P_{3}^{2D}\right\|}_{2},$$
where ${\left\|\cdot \right\|}_{2}$ denotes the Euclidean distance between two points.

According to 3D bounding box model, $P_{1}^{3\mathrm{Di}}$ and $P_{7}^{3\mathrm{Di}}$ are selected, and the diagonal pixel length of 3D bounding box can also be defined as:(12)$$l_{3D}={\left\|P_{1}^{3\mathrm{Di}}-P_{7}^{3\mathrm{Di}}\right\|}_{2},$$

The difference of Equation (11) and (12) consists of a set of diagonal constraint. Figure 8 shows the vehicle diagonal constraint. The red/yellow wireframe represents 2D/3D bounding box. When 2D bounding box and 3D bounding box are completely fitted, the blue line segment indicates that the 2D/3D diagonals completely coincide in the image coordinate system and the value of diagonal constraint is 0, which means 3D vehicle physical size is relatively accurate. The word relatively means the size is accurate in the case of diagonal constraint.

According to the principle of vanishing point, the straight line composed of 0–3, 1–2, 4–7, 5–6 point pairs in the 3D bounding box model must pass the vanishing point along the road direction in the image coordinate system. Therefore, it can be used as another set of constraints to accurately calculate 3D vehicle physical size.

In the image coordinate system, the included angle between two lines (one formed by point pairs, the other formed by one point and the vanishing point along the road direction) can be denoted as θ.

For four point pairs, according to the cosine theorem, we can derive $\theta_{1},\theta_{2},\theta_{3},\theta_{4}$ as follows:(13)$$\cos\theta_{1}=\frac{{\left\|P_{0}^{3\mathrm{Di}}-P_{3}^{3\mathrm{Di}}\right\|}_{2}^{2}+{\left\|P_{0}^{3\mathrm{Di}}-\mathrm{VP}\right\|}_{2}^{2}-{\left\|P_{3}^{3\mathrm{Di}}-\mathrm{VP}\right\|}_{2}^{2}}{2\cdot {\left\|P_{0}^{3\mathrm{Di}}-P_{3}^{3\mathrm{Di}}\right\|}_{2}^{2}\cdot {\left\|P_{0}^{3\mathrm{Di}}-\mathrm{VP}\right\|}_{2}^{2}},$$
(14)$$\cos\theta_{2}=\frac{{\left\|P_{1}^{3\mathrm{Di}}-P_{2}^{3\mathrm{Di}}\right\|}_{2}^{2}+{\left\|P_{1}^{3\mathrm{Di}}-\mathrm{VP}\right\|}_{2}^{2}-{\left\|P_{2}^{3\mathrm{Di}}-\mathrm{VP}\right\|}_{2}^{2}}{2\cdot {\left\|P_{1}^{3\mathrm{Di}}-P_{2}^{3\mathrm{Di}}\right\|}_{2}^{2}\cdot {\left\|P_{1}^{3\mathrm{Di}}-\mathrm{VP}\right\|}_{2}^{2}},$$
(15)$$\cos\theta_{3}=\frac{{\left\|P_{4}^{3\mathrm{Di}}-P_{7}^{3\mathrm{Di}}\right\|}_{2}^{2}+{\left\|P_{4}^{3\mathrm{Di}}-\mathrm{VP}\right\|}_{2}^{2}-{\left\|P_{7}^{3\mathrm{Di}}-\mathrm{VP}\right\|}_{2}^{2}}{2\cdot {\left\|P_{4}^{3\mathrm{Di}}-P_{7}^{3\mathrm{Di}}\right\|}_{2}^{2}\cdot {\left\|P_{4}^{3\mathrm{Di}}-\mathrm{VP}\right\|}_{2}^{2}},$$
(16)$$\cos\theta_{4}=\frac{{\left\|P_{5}^{3\mathrm{Di}}-P_{6}^{3\mathrm{Di}}\right\|}_{2}^{2}+{\left\|P_{5}^{3\mathrm{Di}}-\mathrm{VP}\right\|}_{2}^{2}-{\left\|P_{6}^{3\mathrm{Di}}-\mathrm{VP}\right\|}_{2}^{2}}{2\cdot {\left\|P_{5}^{3\mathrm{Di}}-P_{6}^{3\mathrm{Di}}\right\|}_{2}^{2}\cdot {\left\|P_{5}^{3\mathrm{Di}}-\mathrm{VP}\right\|}_{2}^{2}}$$

The sum of four equations above consists of a set of vanishing point constraint. As shown in Figure 8, the red line segment is used to extract the vanishing point. When 2D bounding box and 3D bounding box are completely fitted, the deep blue line shows that the line formed by point pairs and the vanishing point completely coincide in the image coordinate system and the value of vanishing point constraint is 0, which means 3D vehicle physical size is relatively accurate. The word relatively means the size is accurate in the case of vanishing point constraint.

In this paper, the steps to obtain the vehicle geometric constraints are as follows:
Step 1:YOLOv4 is used to obtain the vertices in the image coordinate system $P_{i}^{2D}=(u_{i}^{2D},v_{i}^{2D}),i=0,2,3$, base point $P_{1}^{2D}=(u_{1}^{2D},v_{1}^{2D})$ and vehicle type.Step 2:$\left(l_{v},w_{v},h_{v}\right)$ is considered to be a set of unknown parameters. The base point in the world coordinate system can be obtained by Equation (8) as $P_{1}^{3D}=(x_{1}^{3D},\;y_{1}^{3D},\;z_{1}^{3D})$, where $z_{1}^{3D}=0$. Then, According to Table 1, the world coordinates $P_{0}^{3D}$, $P_{2}^{3D}$ to $P_{7}^{3D}$ can be calculated.Step 3:According to Equation (11), the diagonal pixel length of 2D bounding box is calculated. Then, the world coordinates of vertex 1 and 7 are converted to the image coordinates according to Equation (7) as $P_{1}^{3\mathrm{Di}},P_{7}^{3\mathrm{Di}}$. Finally, the diagonal pixel length of 3D vehicle bounding box is calculated according to Equation (12), and a set of diagonal constraints are formed.Step 4:According to Equation group (8), the world coordinates of vertices from 0 to 7 are converted to image coordinates as $P_{0}^{3\mathrm{Di}}$ to $P_{7}^{3\mathrm{Di}}$. The values of $\cos\theta_{1}$ to $\cos\theta_{4}$ can be calculated according to Equations (13) to (16), and a set of vanishing point constraints are formed.

According to the above algorithm, the diagonal constraint and vanishing point constraint are obtained to construct the constraint error as $l_{\mathrm{cal}}-l_{\mathrm{truth}}$. Where $l_{\mathrm{cal}}$ is the actual constraint value obtained by calculation, and $l_{\mathrm{truth}}$ is the ideal constraint value when 2D bounding box and 3D bounding box are completely fitted. By analyzing the above algorithm, it can be easily seen that the variables in the constraint error are composed of parameters $l_{v},w_{v},h_{v}$, which can constitute the nonlinear constraint space of parameter vectors.

To sum up, the nonlinear constraint function of the parameter $X=(l_{v},w_{v},h_{v})$ is:(17)$$\arg\underset{X}{\min}\frac{1}{2}(\sum_{i=1}^{N_{f}}\lambda_{d}{\left(l_{2D}-l_{3D}\right)}^{2}+\sum_{j=1}^{4}\lambda_{v}{\left(\cos\theta_{j}-4\right)}^{2}),$$
where $N_{f}$ is the occurrence time of the same vehicle in video frames, $\lambda_{d}$ and $\lambda_{v}$ respectively represent the error coefficient of the diagonal constraint and vanishing point constraint which are usually set to 1 and can be adjusted in different conditions, and $\underset{X}{\min}$ represents the value of X when the constraint function reaches the minimum.

The constraint function is nonlinear. LM (Levenberg-Marquardt) method is adopted in this paper to solve the constraint function, which is easy to reach convergence. The initial value $X_{0}$ can be obtained by referring to the national road vehicle size standard [35] based on the vehicle type derived by YOLO.

After solving accurate 3D vehicle physical size, 3D vehicle detection can be completed. Then, the world coordinates of projection centroids can be calculated. According to Equations (9) and (10), coordinates of vehicles in the panoramic image and other scenes can be obtained. As shown in Figure 9, vehicle spatial distribution and 3D trajectory in cross-camera scenes can be obtained by vehicles in continuous motion.

## 3. Results

In our experiments, we used the Intel Core i7-8700 CPU, NVIDIA 1080Ti GPU (Graphics Processing Unit), 32GB memory, and Windows 10 operating system. The open source framework Darknet is used for vehicle detection.

Experiments are carried out on the public dataset BrnoCompSpeed [36] and actual road scene respectively, and the algorithm illustrated in Section 2 is adopted in the experiments. First, road space fusion algorithm in cross-camera scenes is used to generate the panoramic image of road with spatial information. Secondly, YOLOv4 combined with geometric constraints is used for 3D vehicle detection to obtain projection centroids. Finally, the projection centroids are projected to the panoramic image to derive vehicle spatial distribution and 3D trajectories. The experiments can be divided into the following two aspects: (1) Verify the accuracy of projection centroids obtained by 3D vehicle detection algorithm for vehicle spatial distribution. (2) Compare the proposed 3D vehicle trajectory extraction algorithm with several 3D tracking methods in this paper.

### 3.1. BrnoCompSpeed Dataset Single-Camera Scene

Due to the lack of cross-camera datasets from road surveillance perspectives, we choose a public dataset of single-camera scenes from surveillance perspectives published by researchers of Brno University of Technology for our experiments. The cross-camera dataset made by ourselves and experiments carried out on this scene are described in detail in Section 3.2.

The public dataset BrnoCompSpeed contains six traffic scenes captured by roadside surveillance cameras. Each scene can be divided into left, middle, and right perspectives, with a total of 18 HD (High Definition) videos (about 200 GB). The resolution of all the videos is 1920 × 1080. The dataset contains various types of vehicles such as hatch-back, sedan, SUV, truck and bus, and the position and velocity of vehicles are accurately recorded by radar. Therefore, this dataset can be used to verify the accuracy of vehicle spatial distribution and 3D trajectories in single-camera scenes.

As shown in Figure 10, we select three scenes of different perspectives from six scenes for verification which do not contain winding roads. In all the three scenes, the width of a single lane is 3.5 m, the length of a single short white marking line is 1.5 m, the length of a single long white marking line is 3 m, and the length between the starting points of the long white marking lines is 9 m. First, the three scenes are calibrated separately. Calibration results are shown in Table 2. Based on calibration, the road space fusion algorithm described in Section 2.2.2 is adopted to generate the panoramic image with physical information. Since the scenes in the dataset are single-camera scenes, we generate a roadblock containing physical information for convenience which is shown in Figure 11. Each small square of the roadblock represents the actual road space size of 3.5 × 9 m.

The real position of the vehicle in the world coordinate system is defined as $P_{r}$ and the measured position is $P_{m}$. The effective field of view of the scene is set to $L_{s}$(m). Then, the vehicle spatial distribution error can be defined as:(18)$$error=\frac{||P_{r}-P_{m}|{|}_{2}}{L_{s}/2}\times 100\%,$$

Examples of the vehicle spatial distribution and 3D trajectories in dataset scenes are shown in Figure 12. In this experiment, $L_{s}$ is set to 450 m, and the base point in scene 2 can be selected using either left or right perspective. Each scene contains multiple vehicles, and there are some cases of vehicle occlusion. For each instance, the top image contains 3D vehicle detection and 2D trajectory results, and the roadblock on the bottom side contains vehicle spatial distribution and 3D trajectory results. Each vehicle corresponds to one color without repetition. Table 3, Table 4 and Table 5 correspond to the 3D physical size, the image, and world coordinates and spatial distribution error of each vehicle in dataset scene 1 to scene 3. The value of y-axis in the world coordinate system is presented in an ascending order which indicates the distance between the vehicle and the camera is from near to far. To present the results in a straightforward way, the position and direction of the vehicle is marked in the roadblock with a white line segment and a white arrow respectively.

From the experimental results, it can be seen that the average error of vehicle spatial distribution within the scope of hundred meters is less than 5%, which means the accuracy can reach the centimeter level. In the meanwhile, the proposed algorithm is also adaptable to the situation of part vehicle occlusion.

### 3.2. Actual Road Cross-Camera Scene

To further verify the application ability of the proposed algorithm, we choose the actual road with large traffic flow which is located on the Middle Section of South Second Ring Road in Xi’an, ShaanXi Province, China to make a small dataset of cross-camera scenes. The dataset consists of three groups of HD videos (a total of six videos), and each of which is about 0.5 h long. The resolution of all the videos is 1280 × 720. Figure 13 shows the image of the actual road scenes with no overlapping area which are taken by 2 cameras with a distance of 210 m. In the actual road scene, the road width is 7.5 m, the length of a single white marking line on the road plane is 6m, and the length between the starting points of the white marking lines is 11.80 m and 11.39 m in two scenes respectively. First, the scenes taken by two cameras are calibrated separately. Calibration results are shown in Table 6. Based on calibration, the panoramic image with physical information is generated by the road space fusion algorithm described in Section 2.2.2, which is shown in Figure 14. A degree scale in the image represents an actual distance of the starting points of four white marking lines and 3.75 m in the image width and height direction.

In our experiment, we choose three examples of vehicles, which are shown in Figure 15. For each example (similar to the dataset scene), 3D vehicle detection results in two cameras are shown in the first two lines respectively, and 3D vehicle trajectory extraction results are shown in the third line. Each vehicle corresponds to one color without repetition. Table 7 shows the results of vehicle spatial distribution in actual road scene. Similar to the single-camera scenes, we mark the position and direction of the vehicle in the panoramic image with a green line segment and a white arrow, respectively. From the experimental results, it can be seen that continuous 3D trajectories of vehicles in cross-camera scenes can be effectively extracted.

As shown in Figure 16, the proposed algorithm is compared with the 3D tracking methods based on feature point and 2D bounding box, which are represented by red, green, and orange respectively. It can be seen that the method based on feature point is greatly influenced by vehicle texture and surrounding environment, which cannot reflect true driving direction well, and may not be able to obtain continuous 3D trajectory under the condition of occlusion. The method based on 2D bounding box cannot accurately reflect the true driving position due to an unknown distance from bottom edge to the road plane. The proposed algorithm is superior to the existing methods because it can obtain accurate 3D vehicle bounding box, and is robust to vehicle occlusion and low visual angle of cameras. Comparison of the performance of several 3D tracking methods is summarized in Table 8.

Since the proposed 3D vehicle detection algorithm is based on geometric constraints, the overall processing speed is fast. It can be seen from examples in Figure 15, the average processing speed of our algorithm on the GPU platform is 16 FPS with an average time of 600 ms, which can achieve real-time performance.

During the experiment, it can also be found that the accuracy of vehicle spatial distribution and 3D trajectory is related to the pan angle θ of the camera. Therefore, we count the accuracy under different camera pan angles, which is shown in Figure 17. When the pan angle is close to 0°, the information of the vehicle side surface is invisible, which leads to the decrease of 3D vehicle detection accuracy. In practical applications, the pan angle of the camera can be increased appropriately to retain most of the visual information of the vehicle.

## 4. Conclusions

Through experimental verification, the proposed algorithm of vehicle spatial distribution and 3D trajectory extraction in cross-camera scenes in this paper has achieved good results in both BrnoCompSpeed dataset single-camera scenes and actual road cross-camera scenes. The main contributions of this paper are as follows: (1) A road space fusion algorithm in cross-camera scenes based on camera calibration is proposed to generate the panoramic image with physical information in road space, which can be used to convert multiple cross-camera perspectives into continuous 3D physical space. (2) A 3D vehicle detection algorithm based on geometric constraints is proposed to accurately obtain 3D vehicle projection centroids, which is used to describe vehicle spatial distribution in the panoramic image and to extract 3D trajectories. Compared with existing vehicle tracking methods, continuous 3D trajectories can be obtained in the panoramic image with physical information by 3D projection centroids, which is helpful to applications in large scope road scenes.

However, 3D vehicle projection centroids obtained by the proposed algorithm in this paper is highly dependent on 2D vehicle detection results. When the vehicle is far from the camera, it is prone to be missed of detection and the accuracy will decrease when the camera pan angle is close to 0°. Moreover, the proposed algorithm cannot currently be adapted to various road situations and congested traffic. In future work, a more efficient method for road space fusion can be developed to generate the panoramic image and calculate vehicle spatial distribution more precisely and a more sophisticated vehicle detection network can be designed to fuse various types of geometric constraints to further improve the accuracy of 3D vehicle detection under different camera pan angles. In addition, only straight roads and simple traffic conditions are considered in this paper, which is necessary to be further extended to complex traffic scenes such as road-crossing (containing winding roads) and traffic congestion for more practical and advanced applications. Efforts are also needed to collect a large dataset of these complex traffic scenes for algorithm validation. This direction is a key and difficult point in the future work.

## Figures and Tables

**Figure 1 sensors-20-06517-f001:**
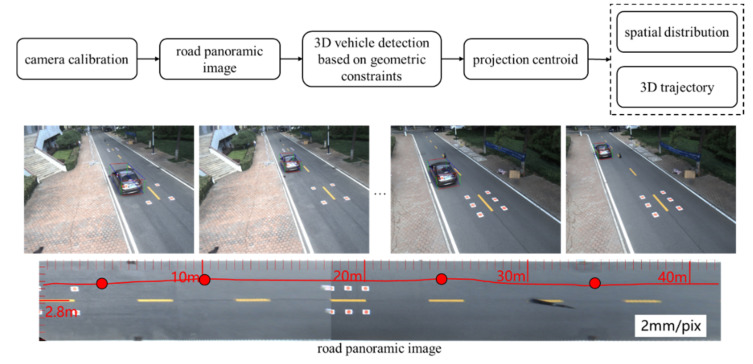
The overall flow chart of the proposed algorithm.

**Figure 2 sensors-20-06517-f002:**
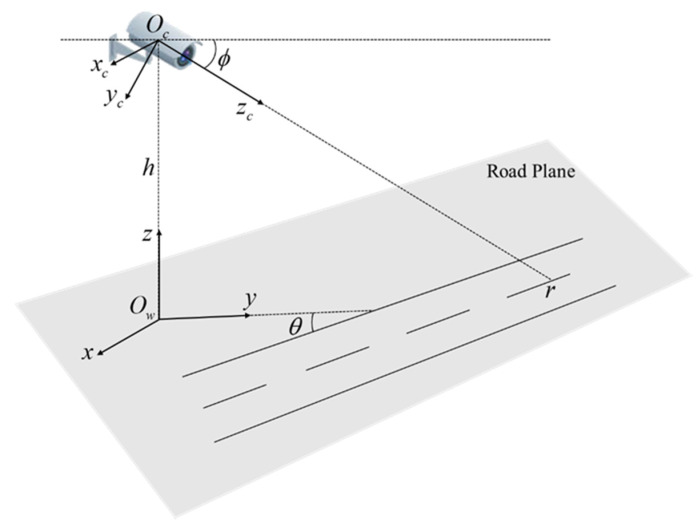
Schematic diagram of coordinate systems and camera calibration model.

**Figure 3 sensors-20-06517-f003:**
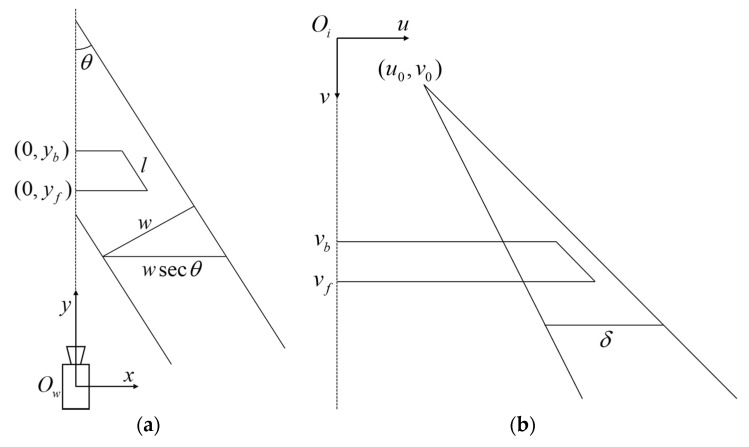
A line segment in the world coordinate system and its projection in the image. (**a**) World coordinate system; (**b**) Image coordinate system.

**Figure 4 sensors-20-06517-f004:**
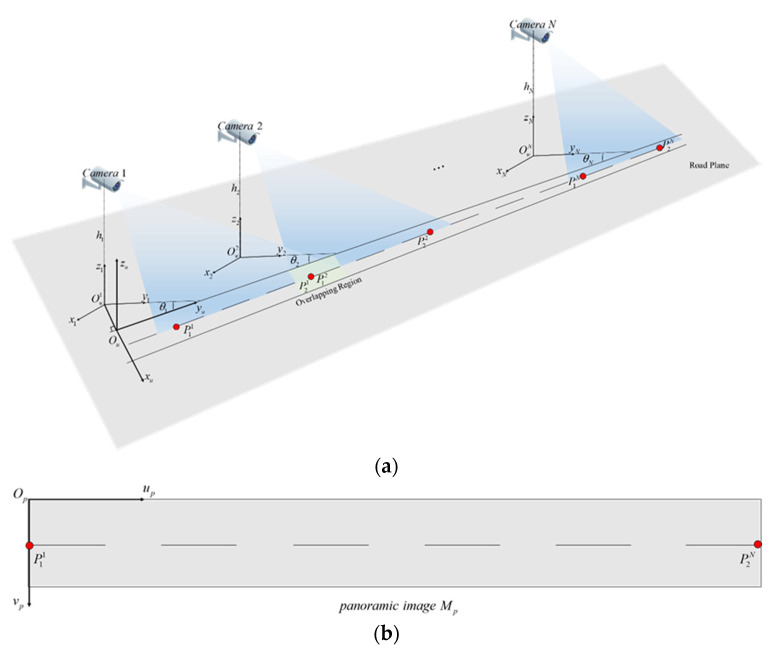
Schematic diagram of road space fusion in cross-camera scenes. (**a**) The unified world coordinate system; (**b**) The panoramic image of road space.

**Figure 5 sensors-20-06517-f005:**
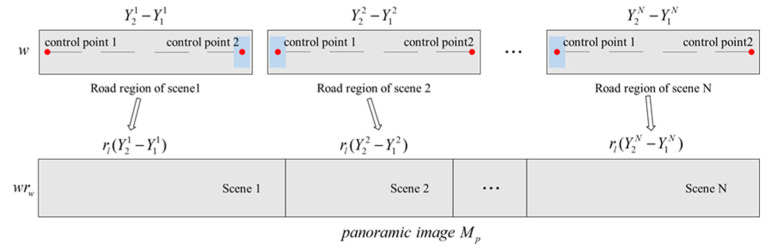
Schematic diagram of road distribution in the panoramic image.

**Figure 6 sensors-20-06517-f006:**
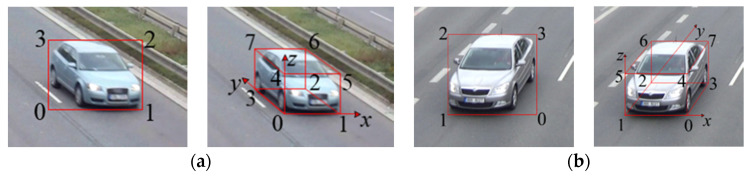
Schematic diagram of vehicle model of 2D/3D bounding box from different perspectives. (**a**) Left perspective; (**b**) Right perspective.

**Figure 7 sensors-20-06517-f007:**
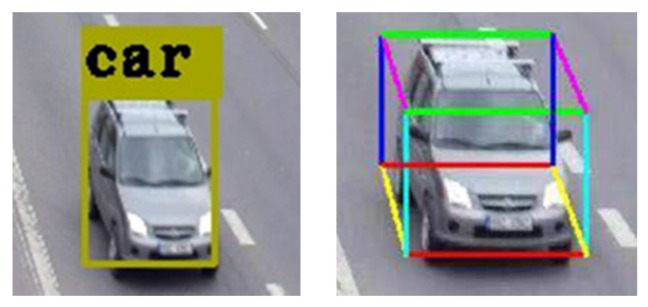
Schematic diagram of 2D (**left**)/3D (**right**) vehicle detection.

**Figure 8 sensors-20-06517-f008:**
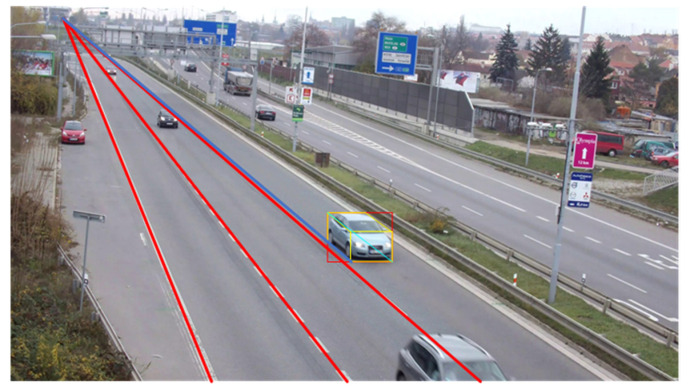
Schematic diagram of geometric constraints.

**Figure 9 sensors-20-06517-f009:**
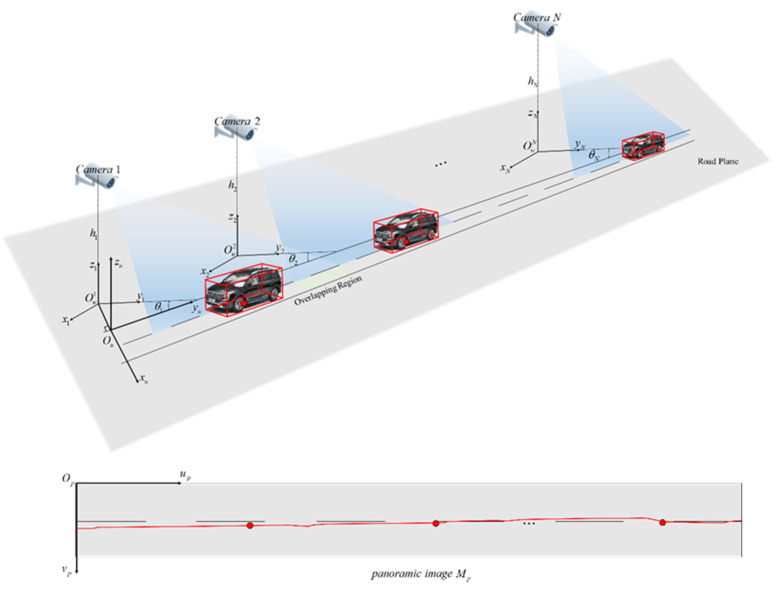
Schematic diagram of vehicle spatial distribution and 3D trajectory extraction in cross-camera scenes.

**Figure 10 sensors-20-06517-f010:**
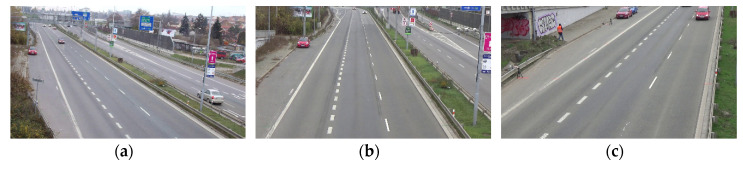
Dataset road scene. (**a**) Scene 1; (**b**) Scene 2; (**c**) Scene 3.

**Figure 11 sensors-20-06517-f011:**
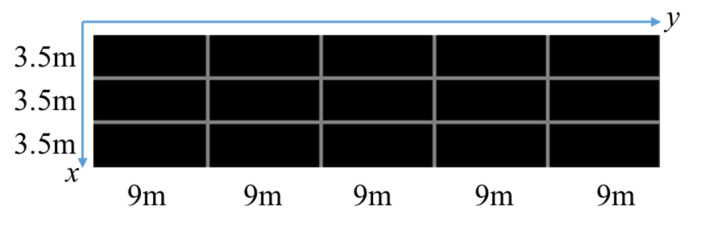
Schematic diagram of the roadblock.

**Figure 12 sensors-20-06517-f012:**
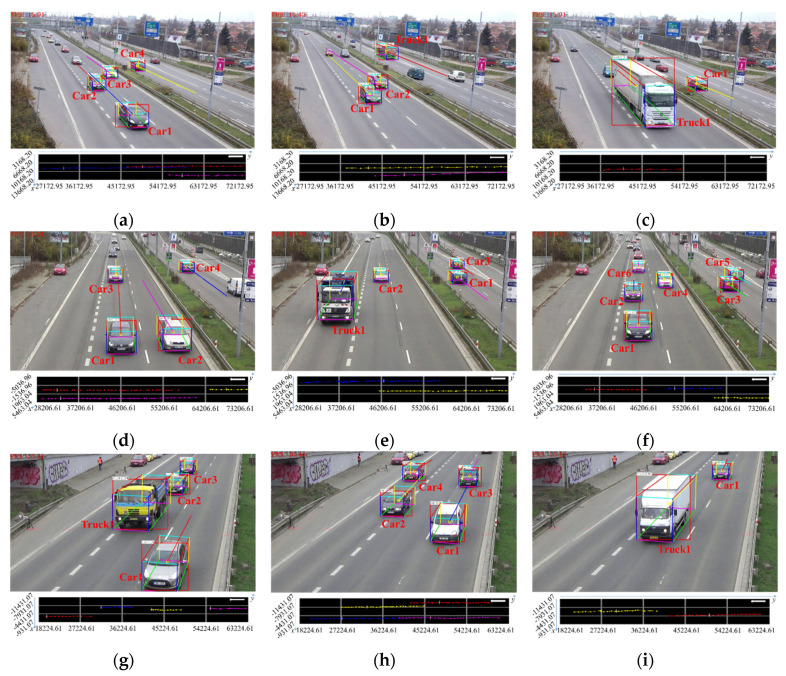
Examples of vehicle spatial distribution and 3D trajectory extraction in dataset road scene. (**a**) Scene1-example1; (**b**) Scene1-example2; (**c**) Scene1-example3; (**d**) Scene2-example1; (**e**) Scene2-example2; (**f**) Scene2-example3; (**g**) Scene3-example1; (**h**) Scene3-example2; (**i**) Scene3-example3.

**Figure 13 sensors-20-06517-f013:**
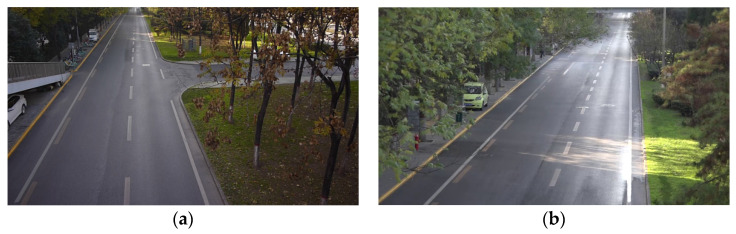
Actual road scene. (**a**) Camera 1; (**b**) Camera 2.

**Figure 14 sensors-20-06517-f014:**
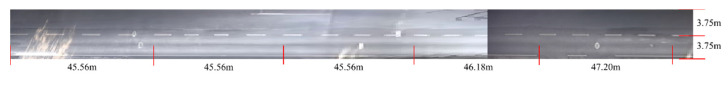
The panoramic image of actual road scene.

**Figure 15 sensors-20-06517-f015:**
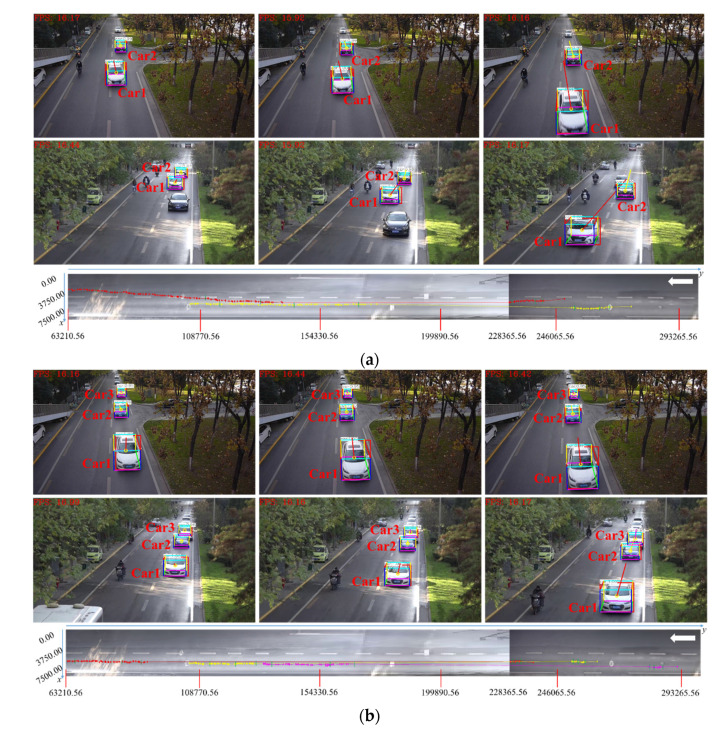
Examples of vehicle spatial distribution and 3D trajectory extraction in actual road scene. (**a**) Example 1; (**b**) Example 2; (**c**) Example 3.

**Figure 16 sensors-20-06517-f016:**
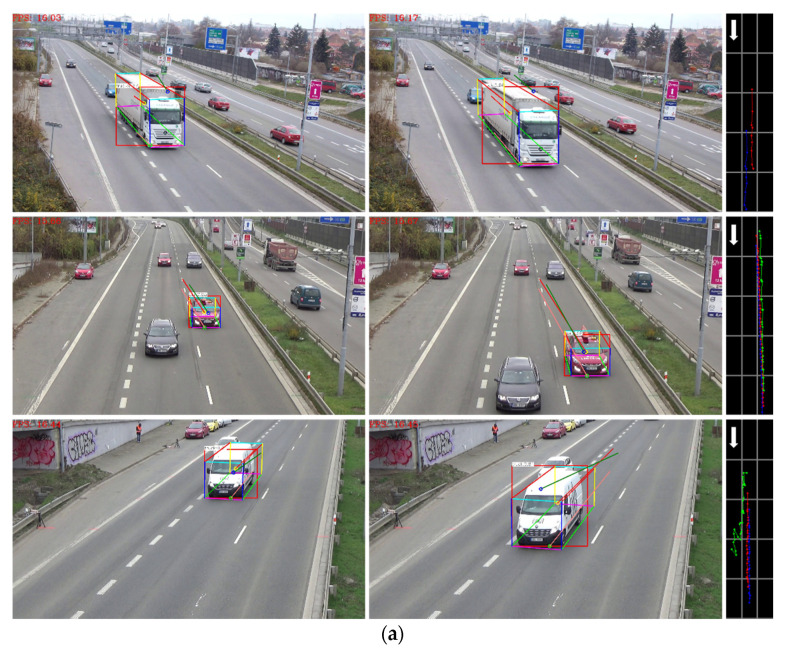
Comparison examples of different 3D trajectory extraction algorithms. (**a**) Comparison example 1; (**b**) Comparison example 2; (**c**) Comparison example 3; (**d**) Comparison example 4.

**Figure 17 sensors-20-06517-f017:**
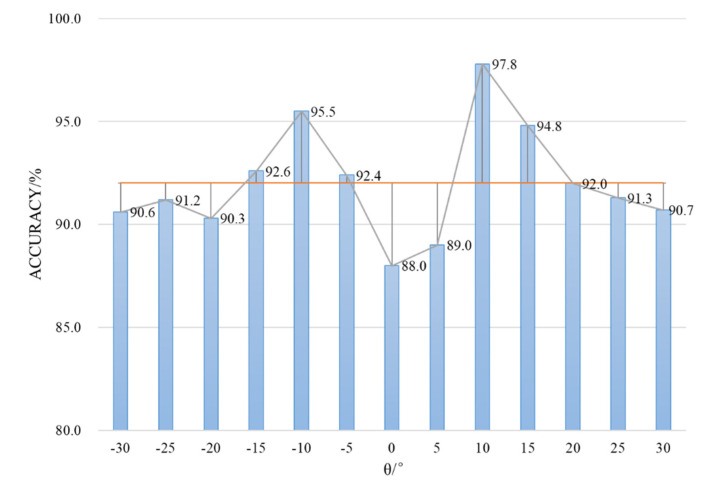
The accuracy of the proposed algorithm with different camera pan angles.

**Table 1 sensors-20-06517-t001:** Eight vertices and projection centroid in 3D bounding box model from different perspectives.

Number	Perspective	
	Left	Right
0	$\left(x_{1}^{3D}-w_{v},y_{1}^{3D},z_{1}^{3D}\right)$	$\left(x_{1}^{3D}+w_{v},y_{1}^{3D},z_{1}^{3D}\right)$
1	$\left(x_{1}^{3D},y_{1}^{3D},z_{1}^{3D}\right)$	$\left(x_{1}^{3D},y_{1}^{3D},z_{1}^{3D}\right)$
2	$\left(x_{1}^{3D},y_{1}^{3D}+l_{v},z_{1}^{3D}\right)$	$\left(x_{1}^{3D},y_{1}^{3D}+l_{v},z_{1}^{3D}\right)$
3	$\left(x_{1}^{3D}-w_{v},y_{1}^{3D}+l_{v},z_{1}^{3D}\right)$	$\left(x_{1}^{3D}+w_{v},y_{1}^{3D}+l_{v},z_{1}^{3D}\right)$
4	$\left(x_{1}^{3D}-w_{v},y_{1}^{3D},z_{1}^{3D}+h_{v}\right)$	$\left(x_{1}^{3D}+w_{v},y_{1}^{3D},z_{1}^{3D}+h_{v}\right)$
5	$\left(x_{1}^{3D},y_{1}^{3D},z_{1}^{3D}+h_{v}\right)$	$\left(x_{1}^{3D},y_{1}^{3D},z_{1}^{3D}+h_{v}\right)$
6	$\left(x_{1}^{3D},y_{1}^{3D}+l_{v},z_{1}^{3D}+h_{v}\right)$	$\left(x_{1}^{3D},y_{1}^{3D}+l_{v},z_{1}^{3D}+h_{v}\right)$
7	$\left(x_{1}^{3D}-w_{v},y_{1}^{3D}+l_{v},z_{1}^{3D}+h_{v}\right)$	$\left(x_{1}^{3D}+w_{v},y_{1}^{3D}+l_{v},z_{1}^{3D}+h_{v}\right)$
Projection centroid	$\left(x_{1}^{3D}-w_{v}/2,y_{1}^{3D}+l_{v}/2,z_{1}^{3D}\right)$	$\left(x_{1}^{3D}+w_{v}/2,y_{1}^{3D}+l_{v}/2,z_{1}^{3D}\right)$

**Table 2 sensors-20-06517-t002:** Camera calibration results of dataset road scene.

	Scene	Scene 1	Scene 2	Scene 3
Parameter				
*f*		2878.13	3994.17	3384.25
*ϕ*/rad		0.17874	0.15717	0.26295
*θ*/rad		0.26604	0.03535	−0.24869
*h*/mm		10119.08	8071.00	8126.49
*VP*		(144.74, 34.78)	(812.62, −109.12)	(1855.68, −373.44)
*H*		$\left[\begin{array}{rrrr} 3025.25 & 154.76 & -170.68 & 1.727\times 10^{6}\\ 5.17 & 18.99 & -2928.28 & 2.963\times 10^{7}\\ 0.26 & 0.95 & -0.18 & 1799.11 \end{array}\right]$	$\left[\begin{array}{rrrr} 4025.18 & 806.43 & -150.27 & 1.213\times 10^{6}\\ -3.25 & -91.80 & -4029.46 & 3.252\times 10^{7}\\ 0.03 & 0.99 & -0.16 & 1263.33 \end{array}\right]$	$\left[\begin{array}{rrrr} 3051.98 & 1731.45 & -249.54 & 2.028\times 10^{6}\\ 88.18 & -347.23 & -3408.29 & 2.770\times 10^{7}\\ -0.24 & 0.94 & -0.26 & 2112.36 \end{array}\right]$

**Table 3 sensors-20-06517-t003:** Measurement of vehicle spatial distribution errors in dataset scene 1.

Instance		Size/m	Image Coordinate	World Coordinate/mm	Error/%
1	Car1	(4.30, 1.80, 1.40)	[963, 861]	[8956.79, 32,729.00, 0]	1.40%
	Car2	(4.30, 1.80, 1.35)	[676, 597]	[8340.02, 49,855.60, 0]	1.85%
	Car3	(4.30, 1.80, 1.35)	[778, 510]	[11,971.37, 58,457.92, 0]	2.54%
	Car4	(4.20, 1.60, 1.35)	[993, 456]	[18,405.90, 64,583.41, 0]	3.66%
2	Car1	(4.40, 1.80, 1.45)	[773, 692]	[8525.12, 42,167.95, 0]	0.86%
	Car2	(4.30, 1.70, 1.40)	[836, 588]	[11,274.48, 49,911.95, 0]	1.61%
	Truck1	(11.00, 2.70, 2.80)	[901, 358]	[21,181.46, 84,556.09, 0]	3.89%
3	Truck1	(20.00, 2.80, 3.80)	[817, 708]	[8950.18, 40,973.38, 0]	2.12%
	Car1	(4.20, 1.60, 1.35)	[1291, 603]	[18,714.95, 46,472.42, 0]	1.39%

**Table 4 sensors-20-06517-t004:** Measurement of vehicle spatial distribution errors in dataset scene 2.

Instance		Size/m	Image Coordinate	World Coordinate/mm	Error/%
1	Car1	(4.50, 1.80, 1.50)	[868, 881]	[382.40, 32,656.53, 0]	1.60%
	Car2	(4.50, 1.80, 1.50)	[1276, 859]	[3892.26, 33,316.97, 0]	1.44%
	Car3	(4.50, 1.90, 1.65)	[828, 380]	[144.69, 68,624.58, 0]	2.80%
	Car4	(4.50, 1.90, 1.55)	[1381, 310]	[11,383.98, 80,370.24, 0]	3.35%
2	Truck1	(11.00, 2.90, 2.80)	[539, 595]	[−3345.10, 46,901.24, 0]	2.03%
	Car1	(4.25, 1.70, 1.35)	[1455, 395]	[10,631.62, 66,104.90, 0]	2.34%
	Car2	(4.30, 1.80, 1.40)	[859, 380]	[679.93, 68,605.65, 0]	1.75%
	Car3	(4.25, 1.70, 1.35)	[1451, 285]	[13,652.22, 85,716.80, 0]	3.84%
3	Car1	(4.40, 1.80, 1.35)	[835, 793]	[120.70, 36,037.78, 0]	1.63%
	Car2	(4.30, 1.70, 1.40)	[798, 515]	[−300.43, 53,117.64, 0]	1.75%
	Car3	(4.30, 1.80, 1.35)	[1539, 453]	[10,753.86, 58,902.55, 0]	2.03%
	Car4	(4.40, 1.70, 1.30)	[1038, 418]	[3486.72, 63,307.65, 0]	3.14%
	Car5	(4.30, 1.60, 1.30)	[1582, 374]	[13,332.64, 69,056.44, 0]	3.16%
	Car6	(4.30, 1.70, 1.40)	[836, 307]	[342.68, 81,376.07, 0]	3.18%

**Table 5 sensors-20-06517-t005:** Measurement of vehicle spatial distribution errors in dataset scene 3.

Instance		Size/m	Image Coordinate	World Coordinate/mm	Error/%
1	Car1	(4.50, 1.70, 1.50)	[1227, 950]	[−3308.45, 19,939.27, 0]	0.31%
	Truck1	(8.90, 2.50, 2.40)	[1054, 479]	[−7100.23, 31,741.25, 0]	1.60%
	Car2	(4.40, 1.60, 1.40)	[1310, 287]	[−6155.65, 42,427.68, 0]	0.57%
	Car3	(4.20, 1.60, 1.40)	[1395, 145]	[−6654.15, 55,028.20, 0]	1.23%
2	Car1	(5.40, 1.90, 1.85)	[1390, 624]	[−3232.59, 27,506.14, 0]	0.72%
	Car2	(4.50, 1.70, 1.50)	[996, 449]	[−7956.65, 32,833.57, 0]	0.93%
	Car3	(5.15, 1.80, 1.70)	[1548, 243]	[−3473.58, 46,422.95, 0]	1.11%
	Car4	(4.30, 1.60, 1.50)	[1120, 202]	[−9853.38, 48,349.16, 0]	1.31%
3	Truck1	(8.90, 2.30, 2.70)	[1074, 545]	[−6371.75, 29,346.67, 0]	1.67%
	Car1	(4.30, 1.70, 1.40)	[1485, 195]	[−4721.74, 50,309.12, 0]	1.72%

**Table 6 sensors-20-06517-t006:** Camera calibration results of actual road scene.

	Camera	Camera 1	Camera 2
Parameter			
*f*		1853.22	5749.81
*ϕ*/rad		0.21361	0.07326
*θ*/rad		0.09411	−0.04820
*h*/mm		7950.72	7877.36
*VP*		(461, −42)	(921, −62)
*H*		$\left[\begin{array}{rrrr} 1903.79 & 448.53 & -135.67 & 1.079\times 10^{6}\\ -3.86 & -40.86 & -1887.41 & 1.501\times 10^{7}\\ 0.092 & 0.97 & -0.21 & 1685.47 \end{array}\right]$	$\left[\begin{array}{rrrr} 5712.37 & 914.59 & -46.85 & 3.690\times 10^{5}\\ 2.98 & -61.76 & -5760.74 & 4.538\times 10^{7}\\ -0.048 & 0.996 & -0.732 & 576.60 \end{array}\right]$

**Table 7 sensors-20-06517-t007:** Vehicle spatial distribution in actual road scene.

Example	Position		World Coordinate in Camera 1/mm	World Coordinate in Camera 2/mm	Unified World Coordinate/mm
1	1-4117	Car1	[182.59, 34,884.74, 0]	not appear in camera 2	[4388.75, 241,500.13, 0]
		Car2	[1214.55, 59,403.27, 0]	not appear in camera 2	[5420.77, 266,018.66, 0]
	1-4122	Car1	[276.22, 31,783.392, 0]	not appear in camera 2	[4482.38, 238,398.79, 0]
		Car2	[1343.44, 55,747.96, 0]	not appear in camera 2	[5549.60, 262,363.36, 0]
	1-4138	Car1	[433.30, 21,957.72, 0]	not appear in camera 2	[4639.462, 228,573.11, 0]
		Car2	[1372.33, 46,263.32, 0]	not appear in camera 2	[5578.49, 252,878.72, 0]
	2-4624	Car1	not appear in camera 1	[−2315.95, 134,591.43, 0]	[4583.15, 134,591.43, 0]
		Car2	not appear in camera 1	[−1860.88, 166,941.60, 0]	[5038.22, 166,941.60, 0]
	2-4660	Car1	not appear in camera 1	[−2921.01, 109,135.85, 0]	[3978.10, 109,135.85, 0]
		Car2	not appear in camera 1	[−1832.96, 146,397.29, 0]	[5066.15, 146,397.29, 0]
	2-4712	Car1	not appear in camera 1	[−4233.11, 70,563.03, 0]	[2666.00, 70,563.03, 0]
		Car2	not appear in camera 1	[−1901.24, 116,527.55, 0]	[4997.87, 116,527.55, 0]
2	1-2801	Car1	[899.46, 26,003.79, 0]	not appear in camera 2	[5105.62, 232,619.19, 0]
		Car2	[1025.87, 49,825.21, 0]	not appear in camera 2	[5232.03, 256,440.60, 0]
		Car3	[1820.65, 78,805.77, 0]	not appear in camera 2	[6026.81, 285,421.17, 0]
	1-2807	Car1	[993.10, 23,872.62, 0]	not appear in camera 2	[5199.26, 230,488.01, 0]
		Car2	[965.72, 47,215.47, 0]	not appear in camera 2	[5171.89, 253,830.87, 0]
		Car3	[1893.67, 76,750.40, 0]	not appear in camera 2	[6099.84, 283,365.80, 0]
	1-2811	Car1	[980.66, 22,200.39, 0]	not appear in camera 2	[5186.82, 228,815.78, 0]
		Car2	[973.85, 45,420.60, 0]	not appear in camera 2	[5180.01, 252,035.99, 0]
		Car3	[1903.34, 75,569.57, 0]	not appear in camera 2	[6109.50, 282,184.96, 0]
	2-3437	Car1	not appear in camera 1	[−1564.61, 91,261.22, 0]	[5334.49, 91,261.22, 0]
		Car2	not appear in camera 1	[−1469.92, 132,265.81, 0]	[5429.19, 132,265.81, 0]
		Car3	not appear in camera 1	[−1300.03, 165,745.23, 0]	[5599.08, 165,745.23, 0]
	2-3455	Car1	not appear in camera 1	[−1687.04, 82,533.43, 0]	[5212.07, 82,533.43, 0]
		Car2	not appear in camera 1	[−1337.24, 126,348.42, 0]	[5561.87, 126,348.42, 0]
		Car3	not appear in camera 1	[−1239.88, 158,211.49, 0]	[5659.23, 158,211.49, 0]
	2-3490	Car1	not appear in camera 1	[−1787.93, 65,696.12, 0]	[5111.17, 65,696.12, 0]
		Car2	not appear in camera 1	[−1548.42, 111,637.28, 0]	[5350.69, 11,1637.28, 0]
		Car3	not appear in camera 1	[−1253.69, 138,779.72, 0]	[5645.42, 138,779.72, 0]
3	1-2588	Car1	[−371.55, 31,699.965, 0]	not appear in camera 2	[3834.615, 238,315.36, 0]
		Bus1	[−1789.39, 69,195.99, 0]	not appear in camera 2	[2416.77, 275,811.39, 0]
	1-2591	Car1	[−399.30, 30,724.42, 0]	not appear in camera 2	[3806.86, 237,339.82, 0]
		Bus1	[−1716.91, 67,609.70, 0]	not appear in camera 2	[2489.25, 274,225.10, 0]
	1-2593	Car1	[−556.38, 29,752.87, 0]	not appear in camera 2	[3649.787, 236,368.26, 0]
		Bus1	[−1738.99, 66,999.57, 0]	not appear in camera 2	[2467.17, 273,614.97, 0]
	2-3151	Car1	not appear in camera 1	[−3437.65, 91,918.145, 0]	[3461.46, 91,918.14, 0]
		Bus1	not appear in camera 1	[−4509.80, 173,877.80, 0]	[2389.30, 173,877.80, 0]
	2-3172	Car1	not appear in camera 1	[−3164.47, 77,067.36, 0]	[3734.63, 77,067.36, 0]
		Bus1	not appear in camera 1	[−4884.50, 168,037.29, 0]	[2014.60, 168,037.29, 0]
	2-3187	Car1	not appear in camera 1	[−3153.01, 67,825.67, 0]	[3746.10, 67,825.67, 0]
		Bus1	not appear in camera 1	[−4941.79, 163,175.51, 0]	[1957.31, 163,175.51, 0]

**Table 8 sensors-20-06517-t008:** Comparison of different 3D trajectory extraction algorithms.

Algorithm		Actual Driving Direction	Actual Driving Position	Continuous 3D Trajectory	Cross-Camera Scene	Panoramic Image
Single-Camera tracking methods	Gu et al. [11]	√	√	√	×	×
	Bullinger et al. [12]	√	√	√	×	×
	Cao et al. [13]	×	×	×	×	×
Cross-Camera tracking methods	Castaneda et al. [26]	×	×	×	√	×
	Peng et al. [28]	√	√	√	√	×
	Qian et al. [30]	×	×	×	√	×
	Ours	√	√	√	√	√

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
