# Peer review of "Vehicle Spatial Distribution and 3D Trajectory Extraction Algorithm in a Cross-Camera Traffic Scene"

_sensors, 2020, doi:10.3390/s20226517_

Round 1

Reviewer 1 Report

The paper proposes an algorithm to obtain three-dimensional trajectory data of vehicles using vehicle spatial distribution algorithm and 3D trajectory extraction. The idea of the paper is sound; however, the motivation and usability of proposed method should be stressed more. For example, changing 2D bounding boxes information to 3D bounding boxes using left and right perspective (What with vehicles from back or turning?). The results do not confirm (several cases presented) that algorithm can effectively complete vehicle spatial distribution and 3D trajectory extraction in various traffic scenes. Several researches based on series of vehicles and at least several testbeds are required. The examples consider one-way multilane roads. What is a usefulness of solution for road crossing and in congested traffic? Finally, the result should be compared against other methods of vehicle tracking (one and multiple camera approach). Some information about sampling should also be added.

Additionally, the related works should be extended by adding:

- several good quality detection methods using one camera,

- brief research on vehicle data fusion from multiple cameras,

- advantage over 2D tracking models. Why a 3D model is better for road solutions (applications for presented straight roads)?

Additionally,

Please define "relatively accurate" statement in line 251

How a percentage error was calibrated in Table 3 (mean of difference?)

Please consider graphical representation of results in tables 3 and 5.

Reviewer 2 Report

This study develops a method of 3D vehicle detection based on geometric constraints. This is an interesting problem. My comments are as follows.

  • The application contexts of the problem should be discussed in the introduction.
  • The overall architect of the proposed framework should be presented. 
  • Please discuss more up-to-date studies regarding the research topic.
  • The time analysis should be included in the experiments. 

Round 2

Reviewer 1 Report

The paper was significantly improved; however, the method should be researcher further (in next papers) to tackle with various road situations and congested traffic. I recommend to publish it with minor changes:

  • line182 small letter -> Where,
  • fig6. add description for every figure (e.g. 2D - 3D)
  • additionally, it would be significant to show pros and cons of the method in summary (e.g. unable to tackle a congested road/ limited road situations) and add some remarks for further researchers - directions of its development.

Reviewer 2 Report

I agreed with the revision.
